# Negotiating extirpation: On the political implications of declaring dugongs extinct in Okinawan waters

## Research Article

Okinawa; Dugong; militarism; protest; extirpation

**Corresponding author:**
Marius Palz;
Email: marius.palz@ikos.uio.no

**Current address:**
Nissan Institute of Japanese Studies, Oxford School of Global and Area Studies, University of Oxford, UK

## Marius Palz [ID]

Faculty of Humanities, Department for Culture Studies and Oriental Languages, University of Oslo, Oslo, Norway

## Abstract

In 2021, scientists published a preprint stating that the dugong population of Okinawa had declined below the minimum viable population and should be considered extinct. The publication led to an outcry amongst Japanese/Okinawan environmentalists and to criticism by international dugong specialists. Two issues were raised: 1) Declaring dugongs extinct, although feeding trails were found in several locations, misrepresented the reality in Okinawan waters, and could have negative impacts on conservation measures; 2) Three authors were sitting on the Environmental Monitoring Committee for a controversial military base construction project in an area where dugongs were frequently spotted before construction commenced. The presence/absence of dugongs at the site had become a political issue, as the animal's protected status and its depiction in folklore gave it symbolic meaning in the anti-base movement. The declaration of dugong extinction reminded protesters of a former Environmental Impact Assessment conducted by Japan's Ministry of Defence, declaring the site to be no relevant dugong habitat. The paper explores the implications of the preprint for the political situation in Okinawa and questions the certainty of dugong extirpation in the region. It argues that speculations about extinction cannot be divorced from the political contexts to which they are invariably tied.

## Impact statement

This article aims to contribute to the fields of extinction studies, environmental anthropology, and Japanese studies during times of increasing environmental crisis. Examining social discourse around the claim of dugong expiration in Okinawa, it provides a case study on how anthropogenic extinction is not just a phenomenon that has to be assessed from the fields of ecology and biology, but one that needs to be also studied from the perspective of the social sciences and humanities. It can therefore be used as an example of how transdisciplinary work between the natural and social sciences can shed new light on one fundamental feature of the Anthropocene. It furthermore functions as a piece of critique towards early declarations of extinction that might have detrimental effects on potential conservation measures.

## Introduction

静かなる　海の宝の　ジュゴンの目
澄んだ瞳で　沖縄みている
*The dugong's eyes, the jewels of the quiet sea*
*With a clear gaze, are looking onto Okinawa*
Opening line from Ito Mitsuru's poem "The Island Where Dugongs Live" (*jugon sumu shima*) (1999: 154), translated by me.

In his poem, Ito Mitsuru describes a dugong observing a place ridden with political conflict around continuous militarism: Okinawa Island in southern Japan. The dugong's clear gaze contrasts with the cloudy view that science has of the remaining individuals living in Okinawan waters. Are the dugongs locally extinct or are they persisting despite massive environmental challenges? The turbidity surrounding these questions is also caused by the very political conflict the poem's dugong has fixed its gaze on.

Dugongs (*Dugong dugon*) are relatives of the manatees, together forming the only surviving species of the Order Sirenia. They are the only herbivorous marine mammals on Earth, depending on seagrass which grows close to the shore (Marsh et al., 2002: 7). Their habitat stretches from coastal waters of the Red Sea and the Indian Ocean to Southeast Asia and the Southwest Pacific. The Okinawan dugongs are the northernmost population of this species. Despite their broad habitat, overfishing, boat collisions, habitat fragmentation, marine pollution, and climate change are putting pressure on the populations. The International Union for



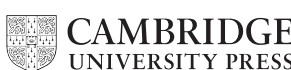

Conservation of Nature (IUCN) categorises them therefore as Vulnerable (Marsh and Sobtzick, 2019). However, the situation of the Okinawan subpopulation is even more severe, standing today on the verge of extirpation. In 2019, the IUCN declared the subpopulation of Japan's southwestern (*nansei*) islands (meaning Okinawa) as Critically Endangered (Brownell et al., 2019).

In the following pages, I will show why it is necessary to analyse claims of potential wildlife extinction in a wider framework that not only builds on the natural sciences but includes the social sciences. I will do so by exploring how Okinawa's critically endangered dugongs are entangled within a net of local, regional, and international politics, American and Japanese militarism, discourses on discrimination towards Okinawa by the Japanese government, and speculations on the extirpation of this charismatic marine mammal.

My argument is that more attention must be paid to the social dimensions of extinction discourses. In the case of Okinawa, different social actors (including different state ministries, scientists, and environmental/anti-military activists) are entangled with dugongs and their potential extinction in various ways that go beyond the mere biological question of whether dugongs are regionally extinct or not. At the core of these entanglements stands the construction of a military base in the waters of Henoko/Ōura Bay, a place in Okinawa's rural north where dugongs were frequently spotted in the past, as well as the animal's cultural significance for the people of Okinawa and its symbolic meaning for the anti-base movement. Anti-base protesters in Okinawa argue that Henoko/Ōura Bay's relevance as a dugong habitat has been understated by Japan's Ministry of Defence, as the presence of these endangered animals at the demarcated base construction site is an obstacle to its militarist development. The existence or non-existence of dugongs has thus become a political issue, shaping human social conflict. For those opposing the base, an ignorance of dugong existence by the Japanese state resembles the colonial mindsets of Japanese and American governments that erased Okinawa's indigenous peoples' relations to land with effects that are still relevant today. The publication of a scientific (preprint) article in 2021 declaring Okinawa's dugongs extinct therefore holds importance in relation to these power structures. For activists, this also means that fighting for recognition of dugongs as prevailing in Okinawa is important not just to implement conservation measures, but also to emphasise the animals' role in the anti-militarist struggle. Dugongs are therefore not just a biologically but also socially relevant species. To build up my argument, I will first introduce my methodology before giving an overview of the topic of militarism and the history of the dugong population decline in Okinawa. Building upon this background information, I will analyse the social meaning of negotiating dugong extinction claims by providing an ethnographic vignette from a meeting between environmentalists and state bureaucrats, before appealing for analysing extinction stories in an interdisciplinary manner to grasp their full meaning.

## Methodology

I conducted 10 months of ethnographic fieldwork in Japan during 2021/22. Staying in a community close to the base construction site, I took part in everyday village life, a method known as participant observation in anthropology. I frequently visited protest events and observed the construction site from protest boats to familiarise myself with the views of the anti-base movement. I conducted forty semi-structured interviews with affected community members, environmental/anti-militarist activists, conservation scientists,

and lawyers and had countless of spontaneous conversations enriching my ethnographic data. Furthermore, I conducted a dugong feeding trail[1] survey together with a group of citizen scientists throughout Okinawa prefecture, familiarising myself with dugong behaviour and seagrass ecology. Literature reviews of historical and secondary sources in Japanese, English, and German were also part of this research.

## Okinawa and militarism

Okinawa is Japan's southernmost prefecture, situated closer to Taiwan than to Tokyo and consists of 54 islands (Okinawa-ken, 2023a). Formerly known as the semi-independent Ryukyu Kingdom (1429–1879) that paid tribute to China and taxes to the Satsuma domain of Japan, it held its own form of governance, until the islands were annexed by the Japanese Empire in 1872. Becoming subjects of expanding Japan, Okinawans entered an ambivalent state. While being exposed to assimilation policies, they remained to be perceived as culturally different and faced severe discrimination. The delay of necessary reforms, suppression of local languages and customs, poverty and resulting mass migration to mainland Japan and foreign countries, and the appointment of mainland Japanese authorities as governors show that Okinawa was treated more like a colony than an equal prefecture until the Pacific War (Tanji, 2006: 24; Meyer, 2020). The Battle of Okinawa (April–June 1945) was one of the last battles of this war, resulting in high numbers of civilian casualties, an almost-complete destruction of the local infrastructure, severe environmental degradation, large-scale displacement of civilians, and further casualties due to hunger and diseases. The experiences of the Battle of Okinawa, told and retold over generations, contribute significantly to contemporary local identity (Allen, 2002: 33–38). After Japan had lost the war, the US held control over Okinawa until 1972, when it was returned to Japan. During this period the US created a vast network of military installations that remains intact until today.

Okinawa Island is the prefecture's largest island, housing most of the prefecture's population (1.3 of 1.5 million people). Okinawa Prefecture also hosts 31 US military facilities that are the workplace of roughly 25,000 American military personnel which make up around 70% of the US Forces Japan. Most of these military facilities are concentrated on Okinawa Island where they occupy roughly 15% of its landmass. Additionally, there are 27 water areas and 20 air spaces designated for US military training (Okinawa Prefectural Government, 2018: 5). This presence comes with severe problems (e.g. illegal land seizure in the 1940s and 50s, environmental degradation, toxic spills, noise pollution, sexual/gender-based violence) which have led to various forms of protest since the end of the Pacific War (Tanji, 2006).

Currently, the construction of a new facility in the waters of Henoko/Ōura Bay is at the focus of opposition against this continuous militarism. Here an existing base is being extended by means of landfill into the surrounding inshore sea to create runways and a port. Constructed by the Japanese government for the American military, its official purpose is to replace the dangerous Marine Corps Air Station Futenma (MCAS Futenma) situated in the middle of densely populated Ginowan City. Protesters see the new facility not just as a replacement but as a strategic upgrade and

---

[1]Characteristic lines in the seagrass bed produced by dugongs during grazing, also known as dugong trenches.

yet another example of American militarism and discrimination by the Japanese government, which is not willing to move the base to the mainland. Although the movement of MCAS Futenma was already decided by the two national governments in 1996, construction only commenced in 2018 due to protests, the filing of several lawsuits, and local anti-base election results (Urashima, 2015; Inoue, 2017; McCormack and Norimatsu, 2018; Nishikawa, 2021; Kumamoto, 2022).

Another reason for opposition is the immense biodiversity of the landfilling site. With over 5,300 confirmed species, including 262 that are endangered according to Japan's national register, Henoko/Ōura Bay is a biodiversity hotspot (Okinawa Prefectural Government, 2018: 18). In this context, one charismatic animal gained especially much attention: the Okinawan dugong.

## A short history of dugong conservation in Okinawa

During times of the Ryukyu Kingdom dugongs were more common and played a vital role in myths and ritual practices. They were depicted in the origin story of humankind recorded on Kouri Island, describing how humans became aware of their own sexuality after witnessing two dugongs mating (Tōyama, 2011: 175). In other legends, they are depicted as human-fish (often translated as mermaids), affiliated with the deity of the sea, calling tsunamis, or warning kind-hearted people of the big waves (Manabe 2002: 51–54; Tōyama, 2011: 176). The islanders of Aragusuku, two small islands in the kingdom's periphery, were the only ones officially allowed to hunt dugongs and pay taxes with their meat, which developed into a delicacy at the court in the capital Shuri (Ikeda, 2012: 152; Tōyama, 2011: 186). With the kingdom's downfall hunting restrictions were abolished and new technologies like dynamite fishing were introduced, leading to a population drop, exacerbated by poverty-driven overfishing in the post-WWII years as well as coastal development work since the 1970s (Tōyama, 2011: 188). In concern about the shrinking population, the dugong was declared a Natural Monument (*tennen kinenbutsu*) under Okinawa's Law for the Protection of Cultural Properties in 1955 and later included in the corresponding Japanese law after the prefecture's reversion in 1972 (Welch et al., 2010: 23–32).

Furthermore, dugongs fall under several domestic conservation laws: the Fishery Resources Conservation Law, the Wildlife Protection and Appropriate Hunting Law, and the Law for the Conservation of Endangered Species. Additionally, the dugong is listed as a species with the highest risk of becoming extinct in the Japan Fishery Agency's Databook of Rare Wild Aquatic Organisms composed in 1998 (Okinawa-ken bunka kankyō-bu shizen hogo-ka, 2008: 11f). It is also listed in Japan's Red List as critically endangered in the wild since 2007, and in Okinawa Prefecture's Red Data Book as critically endangered in the wild since 2005 (Tsuchiya and Adulyanukosol, 2010: 85). On an international level, the Convention on International Trade in Endangered Species of Wild Fauna and Flora (CITES) registered the dugong in 1975 in a list of species with the highest priority of protection (CITES, undated). Japan accepted the convention in 1980. Despite all these laws, listings, and conservation attempts Okinawa's dugongs are on the verge of extirpation, with the IUCN estimating the number of remaining mature individuals to be around 10. Accidental bycatch, coastal development work, and soil runoff remain serious threats to the animals and their habitat in Japan's southwestern islands.

## The environmental impact assessment

The waters of Henoko/Ōura Bay play an important role in the discourse on dugong extirpation in Okinawa. Large seagrass beds and precious potential feeding grounds were situated within the area, parts of which are now destroyed by the progressing base construction. In the past, dugongs were frequently spotted here, which is why they were included in an Environmental Impact Assessment (EIA) conducted by the Okinawa Defence Bureau[2] (ODB) between 2007 and 2009. In the EIA's final version (published December 2012) the ODB concluded that it is unlikely for dugongs to graze on these seagrass beds in the future and that planned measures to mitigate adverse effects on dugongs and their habitat would be sufficient (Okinawa Bōeikyoku, 2012: 6-16-259, 285). The ODB arrived at this conclusion because no dugong activity was detected during their EIA surveys. Activists criticised this interpretation, arguing that dugongs might be avoiding the bay because of noise produced by the ODB's surveys (Tanji and Broudy, 2017: 78). Additionally, during this phase, anti-base protests expanded from land to the inshore sea, where protesters disturbed the ODB's surveys with kayaks and occupied drilling platforms. The presence of ships by protesters, the Coast Guard, and local fishers siding with one site or the other could also be a reason that dugongs were not detected during this period (McCormack and Norimatsu, 2018: 164). After the surveys were finished and water-based protests had calmed down, environmental groups again found feeding trails in the demarcated construction site between 2009 and 2015 (Kasuya and Hosokawa, 2021: 416). However, since drilling surveys and the dumping of concrete blocks for anchoring buoys that demarcate the construction site took place in 2015, no feeding trails have been found.

Two other findings that raise interesting questions considering the relationship between dugongs and the base construction site do, however, exist: In March 2020 an underwater recording device, installed by the ODB, detected high-pitched sounds in the middle of Ōura Bay. Subcontractors of the ODB suggested that these sounds might be dugong calls. Interestingly most of them were recorded during construction pause. This brought into question the ODB's claim that Ōura Bay was not a relevant dugong habitat and that construction activities would not affect the animals (Yoshikawa and Okinawa Environmental Justice Project, 2020: 9). However, upon requests to release the recordings for an independent examination, the ODB refused, claiming that the data was managed by the sub-contracted survey company (Ryukyu Shimpo, 2020). The other finding that holds particular importance for the base construction site is more recent. In July 2022, a citizen found faeces floating in the waters of Kushi Village south of the construction site, where seagrass beds are still intact. A DNA analysis revealed that it belonged to a dugong. However, no animal and no feeding trails were sighted.

In addition, in the last couple of years potential feeding trails were found in various locations throughout the archipelago (Izena/Yanaha, Irabu, Aragusuku, Hateruma, Kuro, Ikema, and Iriomote Islands), some of which correlated with claims of dugong sightings by citizens (Okinawa-ken kankyō-bu shizen hogo-ka, 2022: 5, 12). Another dugong faeces sample was gathered in the waters of Irabu Island in June 2022, some 300 km south of Okinawa Island (Okinawa-ken kankyō-bu shizen hogo-ka, 2023: 38).

---

[2]A subdivision of Japan's Ministry of Defence, who is in charge of the construction of the new base.

As the problematic EIA has shown, legislation and policies for environmental conservation in Japan are easily overwritten by military interests, especially when it comes to the Japan-American security alliance, which forms the basis for the US military's presence in Okinawa and for the construction of the new base.

## At the meeting room

In the summer of 2021, I was invited to join a meeting between environmental activists and Japan's Ministry of Defence and Ministry of the Environment at the House of Councillors Office Building, situated in Tokyo's governmental district. A representative of the Save the Dugong Campaign Centre (SDCC), a Japanese-Okinawan environmental group, welcomed me at the entrance. He, himself Okinawan, reminded me that I should not expect too much from this meeting. He had been to many and had only one way to describe them: frustrating. When we arrived at the meeting room, the activists transformed it into a space of protest, by placing stuffed dugongs in front of them and pinning protest placards to desks and whiteboards. Most of the placards depicted dugongs, stating slogans like "Protect the sea of Henoko, the sea where the dugongs live" (*jugon no sumu henoko no umi o mamorō*), or "We are longing for peace".

Before the meeting started, an indigenous religious practitioner, who is also a prominent figure in the protest movement and a member of the SDCC, zoomed in from Okinawa, emphasising the dugong's meaning as a symbol of peace and an object of worship. After him, two Okinawan delegates from the House of Councillors rose to speak. One of them quoted a newspaper article which mentioned the dugong's critical state nearing extirpation. The other recalled a story from his childhood on one of Okinawa's many islands when his mother had heard that a dugong carcass was found. Back then, she told him a mermaid had died, referring to a common trope in Okinawan legends: dugongs are mermaid-like creatures, half-human half-fish, that are able to converse in human language.

Finally, the SDCC representative who had welcomed me raised one of the main concerns of the meeting: a preliminary scientific report (preprint; not peer-reviewed) published on Research Square, stating that the dugong population of Okinawa has declined below the minimum viable population (MVP), and should therefore be considered extinct. Reconstructing the archipelago's original population by using catch quota from 1893–1920, the preprint's abstract states:

> The initial population size of 300 in the 19$^{th}$ century declined to 50 in 1916 (because of overfishing), 20 in 1979, 10 in 1999, 3 after 2006, and finally extinct in 2019. After 1979, a decline in the natural growth rate for only 20 individuals led to extinction. Long-lived animals fall below the MVP; thus, active conservation measures should have been taken much sooner than when the actual extinction happened. (Kayanne et al., 2021: 2)

The SDCC representative explained, why this article was problematic:

1. Citizens, the Ministry of the Environment, and the Okinawa Prefectural Government had collected evidence that dugongs were still surviving in Okinawan waters, even though direct sightings had become rare. After all, dugong trenches (feeding trails) had been found in several locations throughout the archipelago. Declaring the dugong extinct therefore did not depict the reality in Okinawa correctly and would have negative impacts on conservation measures.

2. Three of the five authors were sitting on the ODB's Environmental Monitoring Committee for the Futenma Airfield Replacement Facility Construction Project, the controversial military base construction project mentioned above. Until 2015 dugong trenches had been frequently found in demarcated construction sites (Kasuya and Hosokawa, 2021: 416).

Once the representatives of the Ministry of Defence had entered the room and introduced themselves, the activists began to confront them with questions. However, the ministry representatives mostly replied by reading out prepared, monotone statements. To the question of whether the Ministry of Defence was involved in the publication of the dugong extinction article by scientists, who were members of the Environmental Monitoring Committee, the ministry officials replied that the article was an outcome of the scientists' individual research activities and that the Ministry of Defence could not comment on it. For the environmentalists, this was not enough. After demanding to elaborate on the Ministry of Defence's standpoint on the current situation of dugongs in Okinawan waters (extinct or not), the ministry's representatives increasingly appeared to be out of words and eventually referred to a declaration of dugongs having an exceptionally high risk of extinction in the wild made by the Ministry of the Environment in 2007. Following up on this, the SDCC's representative asked again: "For clarification, the Ministry of Defence did not ask the members of the Environmental Monitoring Committee to write this kind of article, did it?" (*jijitsu kakunin, bōeishō no hō kara kono kankyō kanshi iinkai no menbā ni konkai no yōna ronbun o kaitekudasai to iu koto wa yōsei shiteinai desu yo ne?*). This type of question was asked several times over and over again in different forms. However, all the Ministry of Defence's representatives did, was read out the ministry's standpoint in a mantra-like manner: the article was composed as an outcome of the scientists' individual research activities (*kenkyūsha kojin toshite no kenkyū katsudō toshite shippitsu sareta mono de aru*) and the Ministry of Defence is not in the position to answer the question and therefore refrains from commenting on it (*bōeishō toshimashite wa kotae no tachiba ni nai kara komento o sashihikaete itadakimasu*). While the ministry's representatives used honorific, yet distant language, the activists sometimes could not hold back their anger, asking questions in highly informal Japanese. Towards the end of the "discussion", the activists handed over a stack of papers, consisting of some 32,000 signatures, demanding the release of the audio recordings of potential dugong calls and a night ban for transport ships.

The following meeting with the Ministry of the Environment was less tense. The activists were less confrontational, and the ministry representatives engaged in a conversation instead of reading prepared statements. The Ministry of the Environment took a very critical stance towards the dugong extinction paper: from their perspective, it was too early to declare dugongs extinct. After all, dugong trenches had been found at several locations throughout Okinawa Prefecture. When asked about the recordings of potential dugong calls in Ōura Bay, the ministry officials only answered that this area falls under the authority of the Ministry of Defence, and therefore they have no insight into the data.[3] Still, the Ministry of

---

[3]While the Ministry of the Environment and Okinawa Prefecture's Nature Conservation Division conduct dugong surveys throughout the prefecture, Henoko/Ōura Bay fall under jurisdiction of the Ministry of Defence, which means that no other authority is allowed to conduct surveys there.

the Environment planned on conducting further surveys throughout the prefecture, despite chronically low funding.[4]

After the meeting, the activists transformed the room from a space of protest back into its former state: protest placards were rolled up and stuffed dugongs were put away. The SDCC's representative seemed tired, and I understood why he had described these meetings as "frustrating." Japan's vertically segmented administrative system (*tatewari gyōsei*) and the refusal by (some) bureaucratic bodies to engage in proper dialogue are hard to deal with when you have an agenda.

## On the uncertainty of extinction and the political power of emptiness

Uncertainty is a fundamental element of extinction. Based on the extinction stories of the Eurasian beaver (*Castor fibre*) in Sweden in the late 19[th] and the thylacine of Australia (*Thylacinus cynocephalus*) in the early 20[th] century, environmental historian Dolly Jørgensen shows how contested knowledge of endangered species populations can be. Ecosystems that are difficult to access, like vast forests or oceans, can especially spark human imagination for being "expansive and inexhaustible" (Jørgensen, 2016: 3). This imagination of abundance, however, can also reverse into speculations about extinction, when sightings of species become rare. The inaccessibility of these ecosystems makes it difficult to detect their inhabitants. Even when scientists conduct surveys, uncertainty can remain. Jørgensen reminds us that

> [environmental] data, in particular, is always based on collection at a certain place at a certain moment in time. Even large sample sets, which may be able to represent the most probable aspects of the environment in question, are not all-inclusive. There are always outliers. Animal populations are particularly problematic to capture because animals are mobile and surveys by necessity are time/space bound. (ibid. 3)

Even though science has progressed in many ways since the times of Jørgensen's examples and new survey methods have been developed, a fundamental issue remains: when species become rare, they become harder to detect. Uncertainty persists around a species' status in a given region if it cannot be detected during a particular survey. This also means that (non-)findings can be interpreted in various ways with real political implications as the ODB's EIA shows. EIAs are tools to identify possible negative effects of development projects on the environment, but they are also "based on collection at a certain place at a certain moment in time." The generated scientific data may turn into "proof" that minimal harm or no harm is to be expected, which allows this tool to become a supporting factor in a project's legitimation (Tanji and Broudy, 2017: 78).

That uncertainty about extinction can hold political importance is not unique to the dugong as for example the controversy over the conservation of the northern spotted owl (*Strix occidentalis caurinain*) in Oregon's old-growth forests has shown. Here conflict between loggers and environmentalists erupted in the late 1980s and early 1990s on the issue of whether forest areas should be protected from logging to save this rare owl species from going extinct. This was feared to have detrimental effects on the logging business putting thousands out of jobs (Satterfield, 2002). Another example of how extinction uncertainty connects to political issues comes from Zanzibar. Here claims of a remaining population of the Zanzibar leopard (*Panthera pardus adersi*) persist, despite an eradication campaign by the local government that started in the 1960s, effectively continuing until the 1990s. Although no scientific confirmation on leopard survival exists on the island, stories about leopards kept by witches and free-roaming ones persist amongst locals (Walsh and Goldman, 2012). Although the examples of the northern spotted owl and the Zanzibar leopard are somehow different from the dugong case, they show us that politics and culture fundamentally influence and are influenced by discourses surrounding rare species and their potential extinctions.

Furthermore, parallels can be drawn between the history of human displacement in Okinawa and declarations of dugong absence in its waters. Examining how the US military frames base sites retrospectively as empty spaces at the point of their construction, geographer Hidefumi Nishiyama shows how ongoing colonialism works by erasing the history of those who originally lived on lands now occupied by these bases. This process is a fundamental feature of imperial and colonial structures that frame conquered and annexed spaces as *terra nullius*, which rationalises the exercise of power over these spaces. In the context of MCAS Futenma, Nishiyama refers to this situation as the "politics of ignorance,"[5] and shows how essential the production of ignorance is, not only to create but also to maintain power:

> The relations between power and ignorance are not unidirectional; ignorance is not just produced by power. Instead, […] the relations between power and ignorance should be understood as mutually constitutive in that the production of a certain type of ignorance contributes to the reproduction of existing power relations. This is not to say that 'ignorance is power'; rather, particular power relations are reproduced through the exercise of ignorance (Nishiyama, 2022: 552).

Nishiyama points out that forgetting, apathy, disinformation, neglect, and secrecy play an important role in the production of ignorance (ibid. 548). From the perspective of Okinawa's protesters, similar mechanisms are at work within the legitimization of environmental destruction in Henoko/Ōura Bay by the Government of Japan. By downplaying the negative impact on the local ecosystem and rationalising its assumptions through scientific data that ignored the area's importance as a dugong habitat, the government attempted to create a public impression that the new base construction cannot be challenged with environmental arguments. Framing the area as empty of dugong existence was an important step in the legitimization of the project. Emptiness or the absence of protected and especially culturally significant species therefore became a powerful tool for the Japanese government in this conflict.

---

[4]Amongst Japanese ministries, the Ministry of the Environment is by far the department with the lowest budget. For the fiscal year 2022, its revenue expenditures were around 321.9 billion yen, while the Ministry of Defence's were approximately 5.3687 trillion yen, 16 times higher (Zaimu-shō, 2022).

[5]I have used a similar term ("political strategy of ignorance") in my article on coral transplantation in connection to the Henoko project in 2021 (Palz, 2021). There, I argue that the Japanese government deliberately ignores scientific recommendations for the transplantation of coral at the construction site to proceed with the project, while maintaining a public façade of doing everything in its power to mitigate adverse effects on the environment. At the same time, it ignores continuing protest in Okinawa, confronting locals with ready-made decisions to present the project as inevitable, influencing how people orient themselves around a future that is shaped by continuous militarism.

## Speculating extirpation

After the dugong extinction paper preprint was criticised by environmental activists in Japan and international researchers, the authors made several changes before eventually publishing the article in Scientific Reports (Kayanne et al., 2022). In the updated version, the possible existence of dugongs around Okinawa's islands is acknowledged in reference to survey outcomes of Okinawa Prefecture and the Ministry of the Environment (which again, were often based on reports of sightings and the findings of dugong trenches by citizens), as well as the recordings of potential dugong calls in Ōura Bay by the ODB (ibid. 6). In addition to the early phase of overexploitation around the turn of the 19th and 20th century, the paper furthermore identifies bycatch in the second half of the 20th century and following genetic deterioration as the main causes for population decline (ibid. 5f). The paper describes the reduction of seagrass beds as not particularly dramatic (from 10,574 km$^2$ in 1979 to 10,497 km$^2$ in 2012, although the source is not clear), thus concluding that "the declining carrying capacity represented by the seagrass bed area was not the major driving factor for the population reduction since 1979" (ibid. 5). Despite these changes, the paper still concludes that the Okinawan dugong population had "crossed the critical extinction risk threshold in 1997" (ibid. 6). In other words, the current remaining individuals fall below the MVP, and therefore, the Okinawan dugong population will become extinct. Whether these conclusions are accurate, is difficult to say. In particular, the importance of seagrass beds might be underestimated, as it is not just the question of how much seagrass remains but also under which conditions. How fragmented are the remaining beds and to what extent do dugongs still have access to them considering high boat traffic in the region? These questions must be investigated before making claims on the interconnectivity between seagrass beds and remaining dugongs.

Furthermore, without denying that the paper identified important points on why Okinawan dugongs are close to extirpation, environmentalists think that it misses a crucial point: the uncertainty that comes with extinction claims. Our knowledge of dugongs in Okinawa relies on sporadic drone, DNA, and feeding trail surveys conducted by chronically underfunded institutions such as the Ministry of the Environment (Kankyō-shō, Not dated) and Okinawa Prefecture's Nature Conservation Division (Okinawa-ken, 2023b). Additionally, reports by citizens give some more information. However, these reports are difficult to verify, as it is not only hard to identify dugongs from a far distance, but unskilled observers might also confuse them with other large animals, such as dolphins or pilot whales. Moreover, fishers pointed out to me during my fieldwork that dugongs in Okinawa only access the inshore sea during the night when boat traffic is reduced. During the day the animals tend to be further out at sea, making it hard to find them. Some researchers also believe that there is a chance for an influx of individuals from the Philippines riding the Kuroshio Current up to Okinawa. DNA testing showed that there are correlations between dugong subpopulations of these two places (Ozawa et al., 2024: 5).

Also, the usage and interpretation of certain terminology affect the discourse on dugongs in Okinawa. In particular, the usage of the term MVP, so the "minimum population size needed for long-term persistence of the species" (Ebenman et al., 2017), seems to have caused confusion. Taking the findings of feeding trails, potential sightings of individuals, and the possibility of an influx of individuals from the Philippines into account shows why. For critics, the question is: is the notion of MVP accurate in this context despite the existence of all this evidence? Resulting in confusion due to the definitional mix-up of concepts like the MVP and a declaration of dugongs having gone (truly) extinct by 2019 as done in the original preprint, creates the risk of either underestimating or overestimating population resilience, depending on political standpoints in the conflict surrounding the base issue.

The inaccessibility of the ocean, the vastness of the Okinawan archipelago, the animal's behaviour, and the fact that surveys are "based on collection at a certain place at a certain moment in time" (Jørgensen, 2016: 3), all give space for speculation whether dugongs will soon become locally extinct or not.

## Conclusion: Embedding extinction stories

Speculations about species extinction are embedded in wider discourses on how humans relate to the environment. In a place like Okinawa, this question is entangled with the controversies of ongoing militarism and the legacies of colonial structures. The discrepancy between the dugong extinction paper preprint and findings by the Ministry of the Environment and Okinawa Prefecture as well as the resulting critique are a good example of how contested extinction can be. To understand this contestation and the objections against the dugong extinction paper, we must look at the wider social relations the dugong's extirpation story is embedded in. These relations include specific types of histories, power imbalances between centre and periphery, and politics of ignorance to exercise power. All of these relations are exemplified by the conflict around the base in Henoko/Ōura Bay, giving the dugong political meaning that transcends its ecological role. Knowledge of what constitutes harm to these animals and whether they are still around at all, therefore becomes capital in this conflict. As the EIA has shown, survey data can be interpreted in a way that supports projects such as the new base. It is therefore not surprising that extinction claims are situated in the longer trajectory of these developments, get politically charged, and indeed contested. It is important to remember that extinction claims are not divorced from political context and can have real life implications not just for the remaining dugongs in Okinawa but also for humans that live there.

This shows that topics that appear to be at the core of the natural sciences are connected to issues that go beyond the scope of these fields. Transdisciplinary work matters, as it is only through employing multiple perspectives that we can make these connections visible. Extinction processes must be assessed from the fields of ecology and biology, but to understand the full meaning of these processes, the social sciences and the humanities must be included. The myriad entanglements between humans and other life forms are constituted by wider social developments but these developments on the other hand are also shaped by what is commonly perceived as "nature". It is not just that anthropogenic species depletion shows us how human behaviour influences non-human others, but also that the presence and absence of species can affect human political discourse, especially in a conflict like the one described above.

What this shows is that the boundaries between "nature" and "culture" are not as solid as they seem to be. This even counts for inaccessible spaces such as the oceans. The social sciences can contribute to making sense of what happens to oceans and their inhabitants in times of mass extinction. As environmental anthropologist C. Anne Claus pointed out, for a long time, fields like anthropology have been mostly terra-focused and were

"surprisingly neglectful of the oceans" (Claus, 2020: 21). This oversight is surely connected to the perception that "the sea is natural, not cultural, and thus devoid of 'signs' of humanity" (ibid. 21). Many scholars have pointed out that human culture cannot be considered as being separate from the natural environment; thus, nature cannot be analysed as being separate from human intervention (Latour and Porter 1993; Ingold, 2000; Haraway, 2003; Hobohm, 2021). The epistemic and systemic nature/culture divide lies at the core of social science's long-time ignorance of the sea. However, the current environmental crisis has shown that even the vastest of oceans are not detached from the effects of human behaviour. Climate change, ocean plastic, and overfishing are only some examples of how deeply interlinked humanity and the oceans actually are. To overcome the nature/culture divide, we must rethink the oceans and stop seeing them as socially empty spaces, because framing something as empty and devoid of multispecies entanglements that include human culture can have severe consequences in this age called the Anthropocene. Looking at case studies like the hopefully avoidable extirpation story of dugongs in Okinawa can help us see these connections.

**Open peer review.** To view the open peer review materials for this article, please visit http://doi.org/10.1017/ext.2024.17.

**Data availability statement.** Data for this article was collected by means of ethnographic fieldwork (participant observation and interviews). It was gathered under consent, agreeing that interview recordings and notes will only be accessed by the researcher and the project leader to protect participants' privacy as common practice in the field of anthropology and as required by the GDPR legislation. Anonymised excerpts from field notes or interviews can be provided by the corresponding author, Marius Palz, upon reasonable request.

**Acknowledgements.** I would like to thank the inhabitants of Okinawa and the representatives of the SDCC who shared their stories and thoughts with me.

**Author contribution.** The author confirms sole responsibility for the following: conception of the study, manuscript preparation and revision.

**Financial support.** Research for this article was conducted in the context of the project "Whales of Power: Aquatic Mammals, Devotional Practices, and Environmental Change in Maritime East Asia." This project is funded by the European Union's Horizon 2020 research and innovation programme under grant agreement no. 803211 (ERC Starting Grant 2018).

**Competing interest statement.** The author declares that he has no conflicts of interest.

**Ethics statement.** Parts of this discussion are also available in the author's PhD thesis.
Palz M (2023) *Searching for Zan: Human-Dugong Relations and Environmental Activism in Okinawa.* Department of Culture Studies and Oriental Languages, Faculty of Humanities, University of Oslo.

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
