## [Editor Report]

Both reviewers find the study topical and relevant. The topic is also very relevant to the Special issue on Extinction across the disciplines. Nonetheless, both reviewers have some concerns. Reviewer 1 raises concerns over the definitions and application of extinction terminology that are fundamental not only to the clarity of the paper, but to geopolitical interpretations. Fundamental to the arguments discussed in the paper are the concepts of extinction and their use by the different actors. Reviewer 2 has concerns over the structure and content of the paper. The focus is of interest and a good fit to the interdisciplinary special issue. However, formatting it as is typical for this journal will strengthen it, and a more in depth analysis of the system (and placing in wider context) is needed. 

It is also important to address reviewer 1’s comments concerning species rarity and detection and this paragraph needs a rigorous summary of the issue in the context of ecology and conservation. 

Reviewer 1 also suggests a number of minor revisions to improve the ms.

Addressing reviewer 1’s comments concerning the definitions of extinction and their use by the different actors is important in the context of this special issue which asks 

what extinction means – biologically, culturally, socially ; and what are the different meanings of extinction in different social and ecological contexts, and what stories and narratives. Furthermore, researching these definitions and their use by different actors, both in the context of the Dugong , but also in the wider context of the literature, and using this to form clear conclusions, will help to address reviewer 2’s concerns about the research component and structure of the paper

---

## [Editor Report]

The authors have addressed the comments made by both of the initial reviewers. In particular they have defined and applied terms around extinction and IUCN categories as recommended and restructuring to include and introduction and methodology whilst retaining a structure appropriate to this interdisciplinary journal. I have a few comments many of which relate to the introduction and scene setting. 

Introduction

This paragraph needs to introduce the background to the study in a manner that can be followed by readers from different disciplines. Currently it assumes that the reader is already aware that plans for a military base would be affected by the need to conserve this species; that dugongs are vulnerable to extinction (IUCN). 

“the existence or non existence” of dugongs- this is the first mention that their status is in debate. 

These points become much clearer on reading the later section on Okinawa and militarism. However, as the lead in to the study, the introductory paragraph needs editing.

Although the conservation status of the dugongs is mentioned, the paper would benefit from a description of international and local conservation regulations this might affect plans to expand the base 

Preprint article 2021- the term preprint indicates that a version is due to be published. On P8 you note that this had not been peer reviewed. I appreciate that the article focuses on the impact of this preprint and am please to see but consideration of the final paper (and whether it differed from the initial) in the discussion. 

“protesters argue” – clarify. People protesting against the military base?

P2 lines 2–3 – forces in Japan?

For the non biologist, explain what is meant by a “feeding trail” earlier in the ms (currently we have to wait to page 8)

P 6 line 9. Additionally, during this phase, anti-base

P9 line 17–18 “the enactment of politics of ignorance…. agenda” is rather unclear (for a biologist, it may be clearer from other disciplines). Ignorance suggests that something is not known about. But the preceding paragraph suggests that the Ministry of defence were arguing that they knew dugongs to be extinct. Later (page 11) you explore this concept in more detail – but ensure that it is briefly defined also at first use.

Bureaucrats- it is not clear if this term refers only to representative from the Ministry of defence, or also includes those from the Ministry of environment. Did the activists also include bureaucrats amongst their number. It feels that this term is being loosely used to imply a lack of engagement. Perhaps describe the language used and statements used rather than the individuals.